# Metagenomic Analysis of the Composition of Microbial Consortia Involved in Spruce Degradation over Time in Białowieża Natural Forest

**DOI:** 10.3390/biom13101466

**Published:** 2023-09-28

**Authors:** Grzegorz Janusz, Andrzej Mazur, Anna Pawlik, Dorota Kołodyńska, Bogdan Jaroszewicz, Anna Marzec-Grządziel, Piotr Koper

**Affiliations:** 1Department of Biochemistry and Biotechnology, Maria Curie-Skłodowska University, Akademicka 19, 20-033 Lublin, Poland; anna.pawlik@mail.umcs.pl; 2Department of Genetics and Microbiology, Maria Curie-Skłodowska University, Akademicka 19, 20-033 Lublin, Poland; andrzej.mazur@mail.umcs.pl (A.M.); piotr.koper@mail.umcs.pl (P.K.); 3Faculty of Chemistry, Maria Curie Skłodowska University, M. Curie Skłodowska Sq. 2, 20-031 Lublin, Poland; dorota.kolodynska@mail.umcs.pl; 4Białowieża Geobotanical Station, Faculty of Biology, University of Warsaw, Sportowa 19, 17-230 Białowieża, Poland; b.jaroszewicz@uw.edu.pl; 5Department of Agriculture Microbiology, Institute of Soil Science and Plant Cultivation, Czartoryskich 8 Str., 24-100 Puławy, Poland; agrzadziel@iung.pulawy.pl

**Keywords:** ATR-FTIR, bacteria, Białowieża, fungi, microbiome, wood

## Abstract

Deadwood plays an important role in forest ecology; its degradation and, therefore, carbon assimilation is carried out by fungi and bacteria. To quantify the abundance and distribution of microbial taxa inhabiting dead spruce logs fallen over a span of 50 years and the soil beneath, we used taxonomic profiling with NGS sequencing of hypervariable DNA fragments of ITS1 and 16S V3-V4, respectively. The analysis of sequencing data revealed a high level of diversity in microbial communities participating in the degradation of spruce logs. Differences in the relative abundance of microbial taxa between the samples of the wood that died in 1974 and 2014, and of the soil in its immediate vicinity, were visible, especially at the genus level. Based on the Lefse analysis significantly higher numbers of classified bacterial taxa were observed in the wood and soil samples from 2014 (wood: 1974-18 and 2014-28 taxa; soil: 1974-8 and 2014-41 taxa) while the number of classified fungal taxa was significantly higher in the wood and soil samples from 1974 (wood: 1974-17 and 2014-9 taxa; soil: 1974-57 and 2014-28 taxa). Most of the bacterial and fungal amplicon sequence variants (ASVs) unique to wood were found in the samples from 1974, while those unique to soil were detected in the samples from 2014. The ATR-FTIR method supported by CHN analysis revealed physicochemical changes in deadwood induced by the activity of fungal and bacterial organisms.

## 1. Introduction

Forests cover only one third of the land surface but contain 80% of the biomass and play a fundamental role in the conservation of over half of the world’s terrestrial biodiversity. Forests deliver many ecosystem services, including carbon stocks and carbon sequestration [1,2]. Forests, and especially unmanaged natural forests, which account for 30% of the global forest area, constitute the most important carbon pools on Earth [3,4,5,6]. A substantial fraction (8%) of the total carbon pool in forests globally is contained within deadwood, i.e., fallen and standing dead trees, branches, and other woody tissues [4]. Decomposing logs are a major long-term source of nutrients and energy in the forest ecosystem. Their decomposition is a crucial process, which connects the carbon pools of forests with the soil and atmosphere [1,6,7], making forests a large but dynamic part of the global carbon cycle.

Wood recycling is key to forest biogeochemical cycles, mainly driven by the microbial community. The decomposition of deadwood is a complex process, during which plant polymers are exploited by microorganisms. Since the deadwood microbiome is dominated by fungi, fungal cell wall constituents represent another important resource, especially in the late phases of decomposition [8,9]. During evolution, fungi turned out to be the most effective deadwood degraders using a large set of extracellular enzymes and low-molecular-weight compounds, called mediators, for the decomposition of recalcitrant plant biopolymers such as cellulose, hemicellulose, and lignin [10]. Fungi also determine the composition of the highly diverse bacterial community [3,11,12]. Recent studies have shown that fungi and bacteria in forest soil overlap in substrate utilization, and the enzymatic systems used by bacteria are highly complex and complementary among taxa [13]. While the bacterial community is composed of low-pH-tolerant bacteria, combining decomposers, commensalists, and possibly mycophages, fungal communities are mainly composed of saprotrophs and parasites [10,11]. Wood-decaying fungi mainly belong to the ecological group of white rot fungi [9,14]. However, the temporal dynamics and the qualitative and quantitative composition of deadwood-associated microorganisms are scarcely analyzed [15]. Recent research revealed the forest microbiome composition in Europe [16,17,18] but did not provide information on the dynamics of the microbiome structure over time in highly biodiverse natural forests.

In order to elucidate the complex relationships and changes taking place in forest ecosystems related to the dynamic pool of carbon stored in deadwood, it is necessary to study the composition and function of the major groups of microorganisms associated with deadwood at different stages of decay. Since natural forests represent an essential ecosystem with high ecological value, including nutrient cycling and the preservation of biodiversity, a better understanding of the ecosystem-level roles played by deadwood-related microorganisms in the forest environment is highly valuable. In this study, we present a metagenomic analysis of complex bacterial and fungal consortia associated with the deadwood of the Norway spruce (*Picea abies* (L.) H. Karst) and surrounding soil in the natural forest of Białowieża National Park (BPN)—one of the best-preserved European temperate forest ecosystems (1).

To reveal potential temporal changes in the composition of microbial consortia (fungi and bacteria) participating in wood degradation, we performed an analysis of the microbiome of spruce logs fallen in 1974 and 2014 (hereafter w1974 and w2014) representing distinct decay stages, and the soil from under the logs (hereafter s1974 and s2014), sampled in 2020. Chemical analyses and AFR-FTIR were applied to assess changes in the wood composition.

## 2. Materials and Methods

### 2.1. Research Plot in Białowieża Forest

The samples were taken from a research plot designed and established in 1964 by Prof. Janusz B. Faliński (Faliński, 1978 [19]) in Białowieża National Park (BPN), north-eastern Poland. The study plot (100 m × 100 m) is located in a mixed deciduous oak–lime–hornbeam forest (*Tilio-Carpinetum*) in forest division 342B in the strictly protected area of BPN (52.740855 N, 23.872432 E) (Figure 1). The plot was internally divided into squares of 10 m × 10 m to facilitate the accurate mapping of standing and fallen trees. In each decade since 1964, all living and dead (standing and downed) trees with a diameter > 15 cm in the thicker end of the trunk were measured and mapped. The existing study design and historical data allow for the sampling of dead wood with a known time since the death of the tree, with an accuracy of ±5 years. The study area is exceptional for its naturalness—it lies in the heart of Białowieża National Park, protecting the most primeval part of Białowieża Forest, which is considered one of the last remnants of the European old growths (Jaroszewicz et al., 2019 [20], Sabatini et al., 2018 [1]). This location guarantees that the direct anthropogenic influences on the interactions between the organisms and their substrates, which were assessed as part of the planned analyses, are minimized. 

### 2.2. Sample Collection

In September 2020, we sampled wood from two spruce logs representing the oldest and the most recently dated decay stages: one from the oldest available cohort of deadwood (registered as a new log on the map from 1974) and one from the youngest cohort (registered as new on the map from 2014) (Figure 1). The spruce was chosen for sampling because logs of this species accounted for nearly 50% of the total number of downed logs on the plot, with the rest representing (in descending order) hornbeam (*Carpinus betulus* L.), small-leaved lime (*Tilia cordata* Mill.), Norway maple (*Acer platanoides* L.), pedunculate oak (*Quercus robur* L.), hazel (*Corylus avellana* L.), and silver birch (*Betula pendula* Roth) [21]. The log fallen in 2014 (hereafter referred to as ‘w2014’) was drilled in the middle of the length from three directions (3 samples) through the surface, sapwood, and heartwood with a sterilized (ethanol flamed) 10 mm drill bit. Sawdust (5 g) was packed into falcon tubes and subsequently stored at 4 °C prior to processing in the laboratory. Rotten wood of the log fallen in 1974 (hereafter referred to as ‘w1974’) was sampled with a sterile spatula (3 samples), packed into falcon tubes, and stored at 4 °C. All the layers (bark, softwood, and hardwood) were mixed together. The closest soil samples (1 m radius) of logs fallen in 1974 (3 samples) and 2014 (3 samples) (hereafter referred to as ‘s1974’ and ‘s2014’, respectively) were taken in triplicate with a laboratory spoon and stored in falcon tubes at 4 °C.

### 2.3. Isolation of DNA and 16S rDNA and ITS1 Sequencing

Prior to nucleic acid extraction, sawdust samples were homogenized to fine powder using a mortar and pestle under liquid nitrogen. Total DNA was extracted from the wood and soil (in total 12 samples) with the Soil DNA Purification Kit (Eurx, Gdańsk, Poland). The DNA quantity and quality were evaluated using a Nanodrop (Thermo Fisher Scientific, Waltham, MA, USA). PCR amplification of targeted regions (highly variable bacterial 16S V3-V4 and fungal ITS1b) was performed using specific primers connected with barcodes. For the bacterial 16S V3-V4 region, the primer sequences were 341F: 5′-CCTAYGGGRBGCASCAG-3′ and 806R: 5′-GGACTACNNGGGTATCTAAT-3′, while for the fungal ITS1 region, the primer sequences were ITS5-1737F: 5′-GGAAGTAAAAGTCGTAACAAGG-3′, and ITS2-2043R: 5′-GCTGCGTTCTTCATCGATGC-3′. The PCR products with a proper size were selected by 2% agarose gel electrophoresis. The same amount of PCR product per sample was pooled, end-repaired, A-tailed, and further ligated with Illumina adapters. Libraries were sequenced on a paired-end Illumina platform to generate 250 bp paired-end raw reads (Appendix A).

### 2.4. Bioinformatic Analysis

Redundancy was removed using the *DADA2* v.1.8 package [22] in R v.3.4.3 (R Core Team, 2016). The parameters for the generation of amplicon sequence variants (ASVs) were as follows: filterAndTrim, maxN = 0, maxEE = 5, truncQ = 2. After dereplication of sequences (derepFastq), exact sequence variants were obtained using *dada*, from which chimeric sequences were removed using *RemoveBimeraDenovo*.

Taxonomy was assigned using the latest version of the RDP database with the Naïve Bayesian Classifier [23]. Mitochondrial and chloroplast sequences classified as non-bacterial were filtered out using the subset_taxa function in the *phyloseq* package [24]. The calculation of alpha diversity indexes, the RDA and LEfSe analyses (LDA score threshold was set to 4 by default), and the graphs were completed in the R v.3.4.3 with *microeco* package (v.0.7.1) [25]. For function annotation of the microbial communities, PICRUSt2 software (Version 2.1.2-b) was used for the analysis of the 16S dataset [26] and FunGuild [27] was employed for the analysis of the ITS dataset. Graphs were prepared in R v.3.4.3 using the *ggplot2* package (v.3.3.5) and the *pheatmap* package (v.1.0.12) [26]. 

### 2.5. Wood Chemical Analysis

Fourier transform infrared (FTIR) spectroscopy is a versatile tool for the characteriztion of soil mineral components, including mineral identification, structural assessment, and the *in situ* monitoring of mineral formation and soil organic matter (SOM) [28,29]. It was also used for the analysis of the expected functional groups in the wood and soil samples. A Cary 630 spectrophotometer with attenuated total reflectance mode (ATR-FTIR) (Agilent Technologies, Santa Clara, CA, USA) was used. FTIR spectra were recorded over 4000–530 cm^−1^. Carbon, hydrogen, and nitrogen (CHN) analyses were carried out using the CHN 2400 elemental analyzer (Perkin-Elmer, Waltham, MA, USA).

## 3. Results

### 3.1. Metagenomic Analysis

We performed fungal and bacterial taxonomic profiling using NGS sequencing of hypervariable DNA fragments of ITS1 and 16S V3-V4, respectively, for both wood and soil samples collected from below the trunks collapsed in the respective years. 

### 3.2. Bacterial and Fungal Community Diversity and Richness Analysis—16S Profiling of Wood and Soil

The analysis of the sequencing data of bacterial 16S DNA extracted from the wood revealed the presence of 105 amplicon sequence variants (ASVs), of which 59 were identified at the genus level, belonging to 43 identified family taxa, 27 order taxa, 17 class taxa, and 5 phyla taxa. The most abundant genera were *Acidibacter* (present in all samples), followed by *Acidisoma* (Figure 2A). *Acetobacteraceae* were the most abundant family taxa. The most abundant phyla comprised Proteobacteria, followed by Acidobacteria, prevailing in both w2014 and w1974 samples (Figure 2B). 

In turn, in the DNA samples extracted from the soil collected near both fallen logs, 112 ASVs were found, of which 79 were identified at the genus level (belonging to 53 family taxa, 31 order taxa, 20 class taxa, and 6 phyla taxa). The most abundant genera were *Roseiarcus*, followed by *Acidibacter* (both present in all samples) (Figure 2C). *Bradyrhizobiaceae* and *Acetobacteraceae* were the most abundant family taxa. At the phylum level, Acidobacteria, Proteobacteria, and Actinobacteria comprised the dominant bacterial taxa (Figure 2D).

Alpha diversity indexes (Chao1 estimator, Shannon index, and Simpson diversity index) calculated on the basis of the 16S rRNA sequencing data showed high bacterial richness and diversity in the wood and soil samples, but no statistical differences were found between the analyzed time points (Appendix A).

Linear discriminant analysis effect size (LefSe) analysis was used to determine potential differences in the microbiome composition of wood and soil samples from different years, and to identify bacterial taxa contributing to the differences. Significantly higher numbers of bacterial taxa were revealed in the w2014 and s2014 samples than w1974 and s1974 (Figure 3A,B). The abundance of the Acidobacteria and Actinobacteria phyla in the w1974 samples and the Acidobacteria phylum in s1974 samples was significantly higher than that in the samples w2014 and s2014. In turn, the Proteobacteria and Bacteroidetes phyla prevailed in the w2014 samples, while the predominance of the Actinobacteria phylum was observed in the s2014 samples (Figure 3A,B). At the genus level, the abundance of bacteria classified as *Caballeronia, Mycobacterium, Paraburkholderia, Mucilaginibacter,* and *Roseiarcus* was higher in the w2014 samples, while the w1974 samples were richer in the *Acidibacter* and *Acidiferrimicrobium* genera. Moreover, substantial amounts of eight ASVs and seven ASVs unclassified at the genus level were observed in the wood samples from 1974 and 2014, respectively (Figure 4A). In the s2014 samples, a higher abundance of *Mycobacterium, Phenylobacterium, Bacillus, Occallatibacter, Acidiferrimicrobium, Roseiarcus, Paraburkholderia, Caballeronia, Streptacidiphilus,* and *Actinospica* bacterial genera was recorded, whereas the s1974 samples were rich in bacteria of the *Acidibacter* genus. Similarly, nine ASVs unclassified at the genus level were abundant in each of the soil samples (s1974 and s2014) (Figure 4B).

A pooled analysis of the wood samples revealed the presence of 49 unique bacterial ASVs, with 34 unique ASVs detected in the samples from 1974, and a core microbiome containing 56 ASVs (Figure 5A). The soil samples contained 44 unique ASVs of bacterial taxa, with 27 found in the samples from 2014, and a core microbiome containing 68 ASVs (Figure 5B).

The functional profiling of the wood and soil bacterial communities performed with PICRUSt revealed the highest abundance of genes coding for putative proteins involved in the metabolism, comprising such processes as transport through membranes (including environmental information processing, and the metabolism of carbohydrates, amino acids, cofactors, vitamins), as well as the processing of genetic information (translation, replication, repair). No functional differences were found in the bacterial communities between the samples of wood and soil from the different time periods (Appendix A).

### 3.3. Fungal Community Composition and Diversity Analysis of Wood and Soil—ITS1 Profiling

Similarly to the metagenomics studies of bacterial diversity, fungal community profiling was performed. The analysis of sequencing data of fungal ITS1 amplicons obtained from DNA isolated from the wood samples revealed, in total, 182 ASVs, of which 120 were identified at the genus level, belonging to 100 family taxa, 50 order taxa, 24 class taxa, and 10 phyla taxa. The most abundant genera were *Hyphodontia*, followed by *Amylocorticium* (both mostly present in the w2014 samples) and *Mortierella* (prevalent in the w1974 samples) (Figure 6A). The most abundant fungal phyla were Basidiomycota (dominating in w2014), followed by Ascomycota (most abundant in w1974) (Figure 6B).

The composition of fungal microbiome in soil samples differed from that in wood, especially at the genus level. Among the 208 detected ASVs, 142 were identified at the genus level (belonging to 99 family taxa, 53 order taxa, 26 class taxa, and 9 phyla taxa), and the most abundant fungal genera were *Rusulla, Mortierella*, and *Lactarius* (present in all soil samples) (Figure 6C). The *Russulaceae* and *Mortierellaceae* family taxa predominated in the soil samples. The most abundant phyla were Basidiomycota, Ascomycota, and Mortierellomycota; however, unlike in the wood samples, their relative abundance was similar in the s1974 and s2014 soil samples (Figure 6D). The alpha diversity indexes calculated on the basis of data from ITS rRNA sequencing of the wood fungal microbiome significantly differed between the w1974 and w2014 samples (Appendix A). The diversity of the w2014 fungal microbiome was lower in comparison to w1974. No significant differences were found in these indexes between the soil samples (Appendix A). The LEfSe analysis revealed a significantly higher number of fungal taxa in the w1974 and s1974 samples (contrary to the bacterial microbiome) (Figure 7A,B). Ascomycota prevailed in samples w1974 and s1974 and Basidiomycota in w2014 and s2014 (Figure 7). The higher abundance of *Botryobasidium* (in the w1974 samples) and *Cladophialophora* and *Trichoderma* (in the w2014 samples) mainly contributed to the differences in the microbiome composition between the wood samples from different years (Figure 8A). Moreover, four ASVs unclassified at the genus level were observed to be abundant in the w1974 samples, and one ASV was abundant in the w2014 samples. The fungal community structure of the s1974 samples was characterized by the higher abundance of the *Rusulla, Humicola, Saitozyma, Leptobacillium, Syncephalis, Cephaloteca, Cladophialophora, Trichoderma, Solicoccozyma, Pochonia, Ilyonectaria, Oidiodendron, Leptodontidium, Penicillum*, and *Paxillus* genera. In turn, the s2014 samples were characterized by high abundance of the *Umbelopsis, Sepedonium, Archaeorhizomyces, Mortierella, Acidiferrimicrobium, Roseiarcus, Paraburkholderia, Caballeronia, Metapochonia, Botryobasidum, Apiotrichum, Lactarius*, and *Talaromyces* genera (Figure 8B). Moreover, high amounts of seven ASVs unclassified at the genus level were observed in the s1974 samples, and four ASVs were abundant in the s2014 samples.

The pooled analysis revealed 144 unique fungal ASVs in the wood samples, with 117 present in the w1974 samples, and the core microbiome was represented by 38 ASVs. In the soil, 109 unique fungal ASVs were detected, including 68 ASVs from s2014, and the core soil microbiome comprised 99 ASVs (Figure 9).

The functional profiling of the fungal microbiome in the wood and soil samples with FunGuild revealed the presence of all three main trophic modes in our dataset (Figure 10). The w2014 samples were strongly dominated by fungi belonging to saprotrophic guilds, whereas symbiotrophs and saprotrophs–symbiotrophs prevailed in the w1974 samples. Among the fungal communities in the soil, symbiotrophs prevailed in both s1974 and s2014 samples (Figure 10B). 

### 3.4. Chemical Composition of Soil and Wood Samples

As part of the analysis of the deadwood, the interpretation of ATR-FTIR spectra allowed for the identification of not only a linkage between structural elements in the polymer matrix of the wood but also changes in the deadwood caused by microbial activity and alterations induced during the aging time. The FTIR spectroscopy revealed that bands representing minerals at 3338 cm^−1^, 1668 cm^−1^, and 808 cm^−1^ were assigned to the presence of Al-OH, quartz Si-O at 1921 cm^−1^, and Si-O at 778 cm^−1^. Moreover, there were stretching vibrations of structural hydroxyl groups (OH) between 3750 cm^−1^ and 3400 cm^−1^ and bending bands located between 950 and 600 cm^−1^. The ratio of band intensity or the area for aliphatic stretching at 3000–2800 cm^−1^ relative to aromatic C=C and carbonyl C=O stretching at 1740–1600 cm^−1^ had an impact on soil hydrophobicity, which corresponds to soil wettability. The ratio of aliphatic C-H stretching at 3000–2800 cm^−1^ relative to C=C and/or C=O stretching at 1740–1660 cm^−1^ was used as an index of SOM humification. The comparison of both peaks revealed a decrease in the intensity of the band at 3338 cm^−1^ related to the lower water content (H-bonded OH) and at the band at 1668 cm^−1^ in the case of the s2014 sample. The medium intensity band at 1265 cm^−1^ indicated the presence of soil humic and fulvic acids as salts in the s1974 sample. The lower-intensity band at 1045 cm^−1^ can additionally be assigned to the Si-O vibrations of clay minerals (Figure 11).

Two absorption signal areas, which indicate the hydrophobic (CH-groups) and the hydrophilic (CO-groups) groups, were recorded. The C-H bands occurred at 3000–2800 cm^−1^, while the C-O bands were observed at 1740–1600 cm^−1^. It was found that bands at 3375 cm^−1^, 2940 cm^−1^, 1721 cm^−1^, 1592 cm^−1^, 1506 cm^−1^, 1458 cm^−1^, and 1417 cm^−1^, as well as 1265 cm^−1^, 1213 cm^−1^, 1137 cm^−1^, and 1045 cm^−1^, were related to the presence of lignin (C–O, C=O, CH_2_), cellulose (CH_2_), and hydroxyl groups associated with the hygroscopic properties of wood and deadwood (Figure 11B). In the ATR-FTIR spectra, the bands for the w1974 sample in the range of 3000–3700 cm^−1^ corresponding to absorbed water and –OH groups had evidently lower peaks, which was associated with aging. The broad peaks observed at 3338 cm^−1^ (Figure 11A) and at 3339 cm^−1^ (Figure 11B) can be deconvoluted into bands linked to different wood polymers and involved in the uptake of moisture and assigned to weakly absorbed water, both of which were potentially higher in the wood samples from 1974. The bands at 2940 cm^−1^ and 2894 cm^−1^ may be assigned to the asymmetric stretching vibrations of C-H related to methyl and methylene lignin, cellulose, and hemicellulose. The band at 2840 cm^−1^ was assigned to symmetric CH_2_ stretching vibrations. This band was shifted to a lower wavenumber. The next bands, at 1721 cm^−1^, 1651 cm^−1^, and 1592 cm^−1^, were assigned to the C=O stretching of acetyl or carboxylic acid groups in xyloglucan, bending of H-OH groups, and aromatic skeletal vibrations and C=O stretching of the lignin aromatic ring. The bands at 1506 cm^−1^, 1458 cm^−1^, and 1417 cm^−1^ were assigned to C=C stretching of the aromatic ring, asymmetric bending in CH_3_ (lignin), and CH_2_ bending (cellulose). In turn, the bands observed at 1265 cm^−1^, 1213 cm^−1^, and 1137 cm^−1^ were assigned to guaiacyl ring vibrations and C-O stretching in lignin, C-O stretching in ester groups in lignin, and asymmetric stretching C-O-C in cellulose. The bands at 1045 cm^−1^, 858 cm^−1^, 809 cm^−1^, and 769 cm^−1^ were assigned to stretching in cellulose and non-cellulosic polysaccharides, out-of-phase asymmetric ring stretching (cellulose), the in-plane symmetric vibration of C-H in cellulose, and C-H out-of-plane bonding at all lignin H-unit positions. 

### 3.5. Analysis of Carbon, Hydrogen, and Nitrogen Content (CHN) in Wood and Soil Samples

The chemical composition of the lignocellulosic material obtained from the wood samples was analyzed using the CHN method, which revealed that the w1974 samples contained close to four times more nitrogen than w2014. The carbon content was close to three times higher in s1974 than in s2014, and the hydrogen content in the s1974 soil samples was twice as high as that in the s2014 samples (Table 1). 

## 4. Discussion

While a general model of microbial wood degradation is studied in detail, the succession of microorganisms during decomposition remains unpredictable, mainly due to the long timeline of the processes and numerous environmental factors that should be taken into account (e.g., climate, biodiversity, humidity of substrate, contact of deadwood with soil). Microbial taxa differ in the capacity and efficiency of wood catabolism; therefore, elucidation of the natural dynamics of fungal and bacterial communities’ succession during the decomposition of wood can help to describe some of these functional changes [30,31]. The research plot designed and established in 1964 by Prof. Janusz B. Faliński (Faliński, 1978 [19]) in Białowieża National Park (BPN), in which every dead and fallen tree was registered and described for almost sixty years, provides an unique opportunity to study the microbial succession of wood degradation over a span of years. The study of the microbial community engaged in the degradation of spruce logs fallen in 1974 and 2014 showed significant differences in composition at every taxon level, especially at the genus level. In the wood samples from 1974 and the soil from 2014, a higher number of bacterial taxa were observed. Microbial growth in the soil near the log that fell in 1974 may be limited due to the presence of recalcitrant/toxic lignin, which was also confirmed by the ATR-FTIR and CHN analyses revealing higher carbon, hydrogen and nitrogen concentrations in s1974 in comparison to s2014, which may be result of lignin migration from wood to soil. The w2014 sample was dominated by Basidiomycota phylum fungi, which are known structural wood decayers, dominating during the early stages of wood decomposition [32]. Basidiomycota limit the growth of bacteria [33], and this may have been the cause of the lower bacterial diversity in fresh wood samples. The functional analysis performed in this study revealed that the w2014 sample was dominated by saprotrophs, whereas w1974 was dominated by symbiotrophs/saprotrophs, with a tendency towards the prevalence of symbiotrophs. This was an expected result, as six-year-old deadwood still contains many nutrients and is still in a rather early stage of the decomposition process, carried out by the saprotrophic organisms [32]. The soil from both time points was mainly colonized by symbiotrophs; the s2014 sample was enriched with saprotrophs probably resulting from the log decay process and fungal mycelia of saprotrophs exploring the neighbourhood of the log in search of nitrogen resources [34]. As reported by Kielak et al. [35], the shift in bacterial communities in response to progressive decay is reflected by a positive correlation between bacterial richness and diversity versus the stage of wood decay. They also suggested that wood-inhabiting N-fixing bacteria may support fungi in fulfilling their nitrogen requirements, which was observed in our CHN analysis showing that the nitrogen content was three times higher in the w1974 sample than in w2014, and slightly higher in s1974 than in s2014. These data might be a result of the higher abundance of microorganisms; moreover, the presence of taxa of the *Rhizobiales* class capable of nitrogen fixation was confirmed in the w1974 sample. Bacteria may act as secondary wood colonizers, not only through their nitrogen fixation activity but also through modification of the wood composition by enzymes and radicals, thereby providing nutrients for other organisms [36]. Actinobacteria are well-known as degraders of plant material in soil, similar to saprotrophic fungi, and are therefore strong competitors for root litter, whereas Acidobacteria are able to grow in the low-pH environment of decaying wood [37]. Both bacterial taxa were found in the w1974 sample. Bacteroides (predominant in w2014) are known as litter-associated bacteria with a broad substrate range [38]. The w2014 samples contained bacteria known to possess lignin-degrading capacities, i.e., *Burkholderiaceae* and *Sphingobacteriacease* [39]. As suggested by Zhang, Liu, Han, Zhang, Zhang, He, Li and Cao [36], bacteria mainly prefer the r-strategy (colonizers), whereas fungi adopt the k-strategy (competitors). The former strategy is predominant in early successional forests, whereas the latter is widespread in later successional forests. Interestingly, the abundance of fungal taxa was over four times higher in in the w1974 sample in comparison to w2014, with a large number of common taxa. The wood samples from 2014 were dominated by *Hymenochaetales* fungi, comprising many species from the ecological group of white rot fungi capable of the enzymatic degradation of wood. On the other hand, another fungal order, *Chaetothyriales,* which is abundant in these samples, is known to cause soft rot [40]. However, the w1974 sample was mainly inhabited by fungi belonging to other orders of *Helotiales* and capable of wood degradation, also living in the soil as symbiotrophs [41,42]. The activity of bacterial/fungal organisms caused spectral changes, particularly in lignin (CH, C–O, C=O, CH_2_), guaiacyl, and hemicellulose (CH, C=O, CH_2_). The reduced intensity of the band at 1740 cm^−1^ was related to the increased number of acetyl groups. It is well-known that the main difference between softwoods and hardwoods is the large number of methoxyl groups found in hardwoods assigned to the band at 1600 cm^−1^ [43]. Softwood lignins are made up of guaiacyl units, which are related to the range of 1592–1956 cm^−1^, while hardwood lignins contain a mixture of guaiacyl and syringyl units (the doublet can be detected at the 1610–1596 cm^−1^ range). In this study, only one band was detected (1595 cm^−1^); however, its intensity strongly decreased after 50 years of aging (sample w1974). It is noteworthy that, through a comparison of the intensity of the bands at 1592 (1956) cm^−1^ and 1510 (1506) cm^−1^, the content of syringyl and guaiacyl units (trans-coniferyl alcohol) can be estimated [44], which allows for the conclusion that lignin in the w1974 sample was decomposed. It can be concluded that the ATR-FTIR method revealed physicochemical changes in the deadwood induced by fungal and bacterial organisms. The bands related to the lignin component (1592 cm^−1^, 1510 cm^−1^, 1465 cm^−1^) in the w1974 sample significantly decreased due to the microbial activity. The sharp band at 1022 cm^−1^, assigned to the cellulose unit, partially disappeared, which means that the polymer structure was degraded to some extent. 

The natural forest of Białowieża National Park (BPN)—one of the best-preserved European temperate forest ecosystems—is the ideal place to reveal potential temporal changes in the composition of microbial consortia (fungi and bacteria) participating in wood degradation and related changes in lignocellulose chemical composition. The obtained results clearly indicate that wood degradation is a highly complex process, realized by a consortia of organisms (both fungi and bacteria), whose qualitative and quantitative composition is changing over time. The engagement of dozens of fungal and bacteria species in wood degradation process is implied to involve the activity of numerous enzymes and low-molecular compounds. The next research step should be to describe functional enzymes engaged in the lignocellulose degradation pathway over time.

## Figures and Tables

**Figure 1 biomolecules-13-01466-f001:**
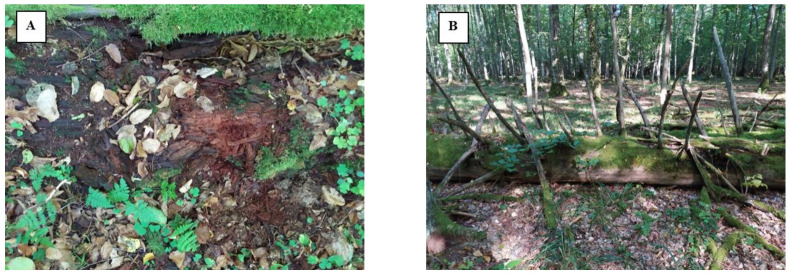
The area plot and logs fallen in 1974 (**A**) and 2014 (**B**), from which the samples were taken.

**Figure 2 biomolecules-13-01466-f002:**
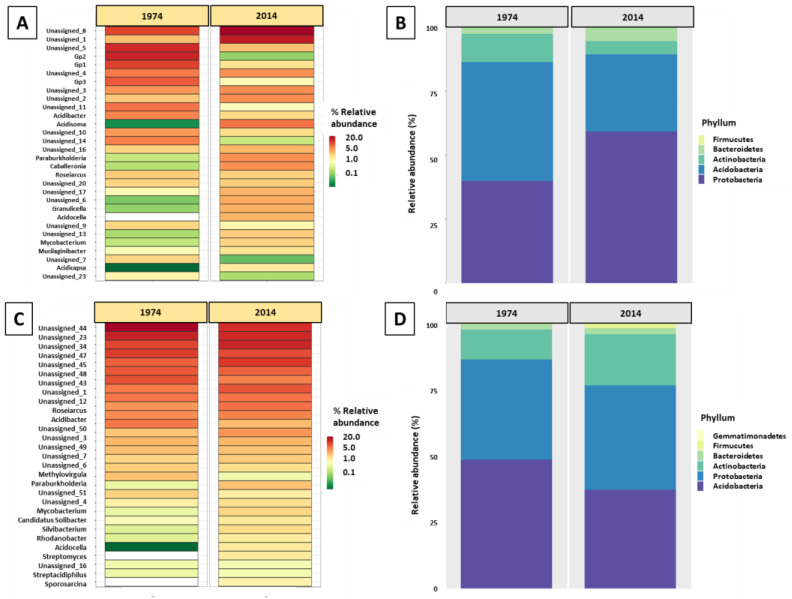
Composition of bacterial genus (**A**,**C**) and phylum (**B**,**D**) taxa based on the metataxonomic analysis of 16S rRNA gene amplicons of wood (**A**,**B**) and soil (**C**,**D**) samples from the logs fallen in 1974 and 2014.

**Figure 3 biomolecules-13-01466-f003:**
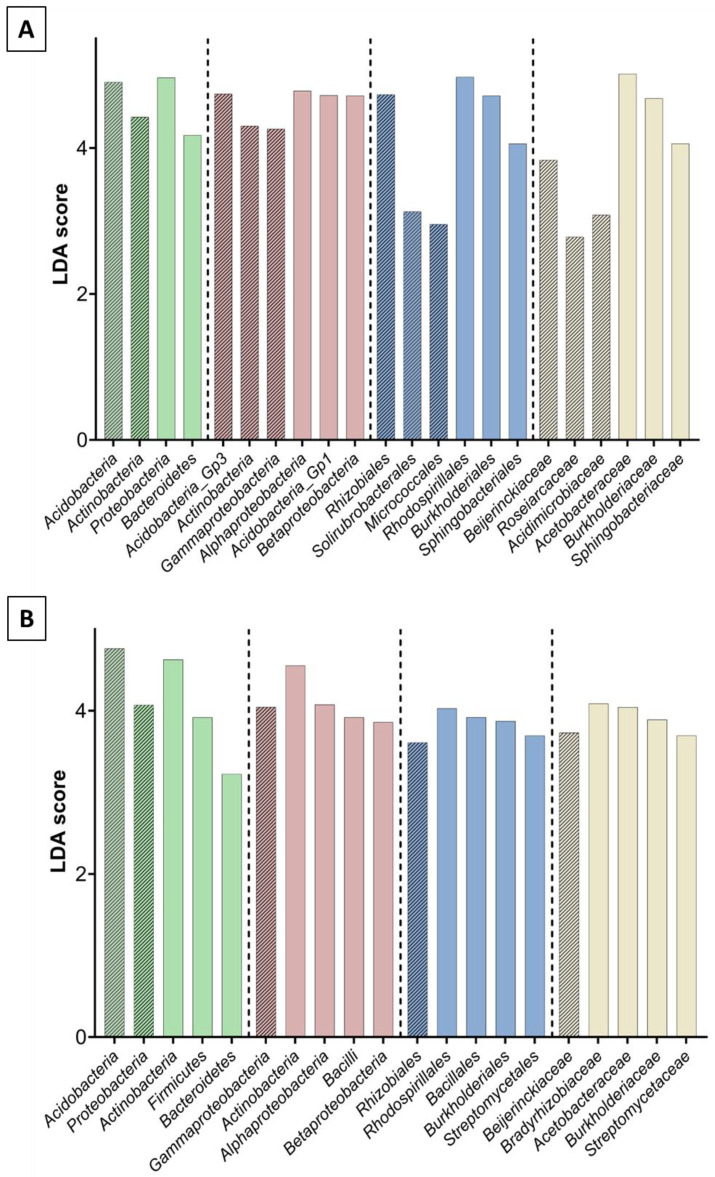
LEfSe analysis showing differences between the metataxonomic results of wood (**A**) and soil (**B**) samples from 1974 (hatched) and 2014 (plain): LDA score representing differences in the abundance of bacterial taxa at the level of phylum (green), class (red), order (blue), and family (yellow).

**Figure 4 biomolecules-13-01466-f004:**
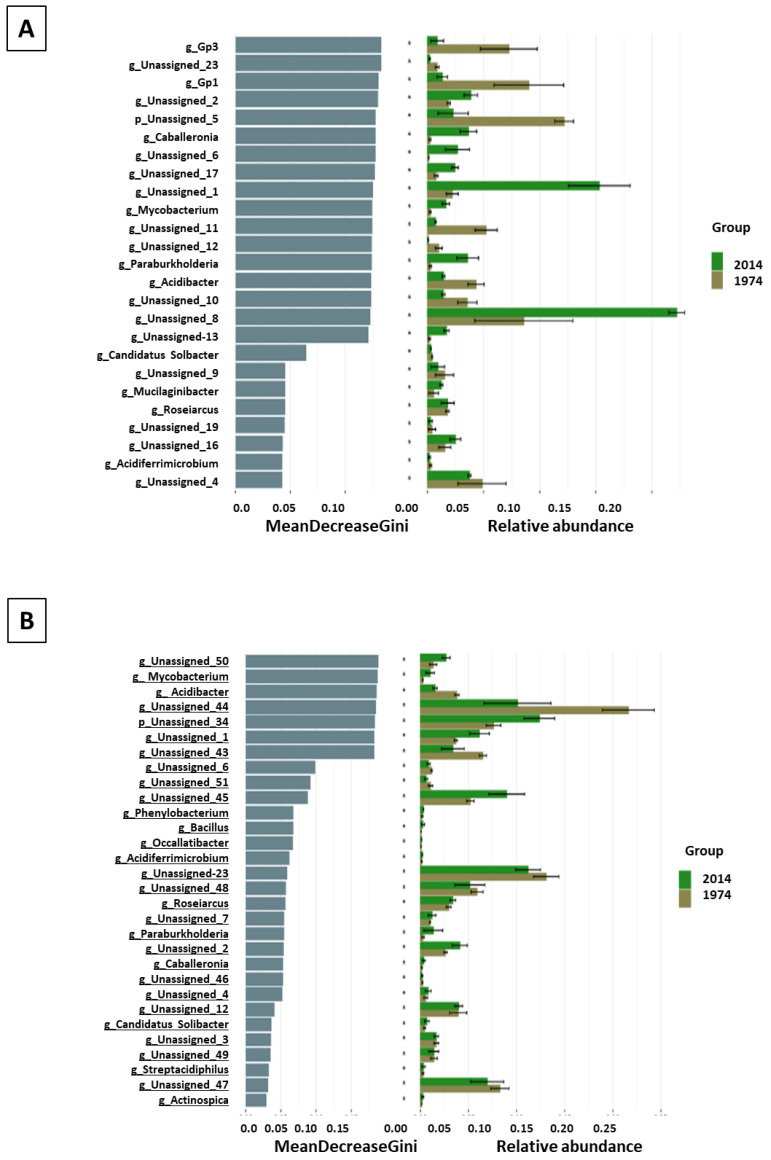
LEfSe analysis showing differences between the metataxonomic results of wood (**A**) and soil (**B**) samples from 1974 and 2014: rf analysis (random forest) showing differences in the abundance of bacterial taxa at the genus level.

**Figure 5 biomolecules-13-01466-f005:**
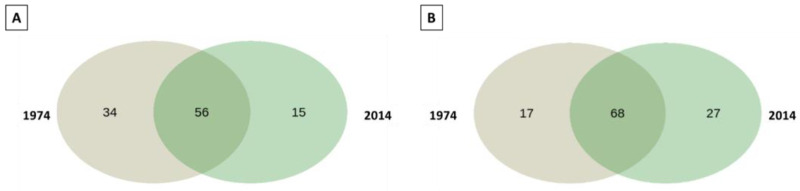
Venn diagram prepared using the 16S rRNA gene sequencing of wood samples (**A**) and soil samples (**B**) from 1974 and 2014. Data presented in the graph refer to the unique, age-specific ASVs and a core microbiome.

**Figure 6 biomolecules-13-01466-f006:**
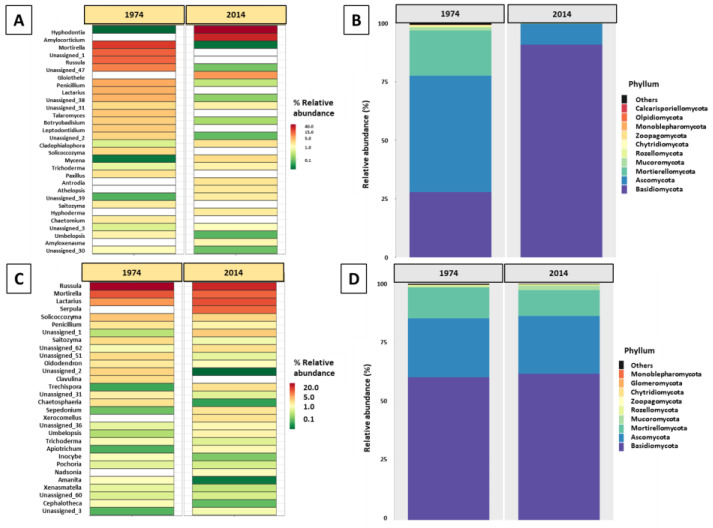
Composition of fungal genus (**A**,**C**) and phylum (**B**,**D**) taxa based on the metataxonomic analysis of ITS1 amplicons of wood (**A**,**B**) and soil (**C**,**D**) samples from under the logs fallen in 1974 and 2014.

**Figure 7 biomolecules-13-01466-f007:**
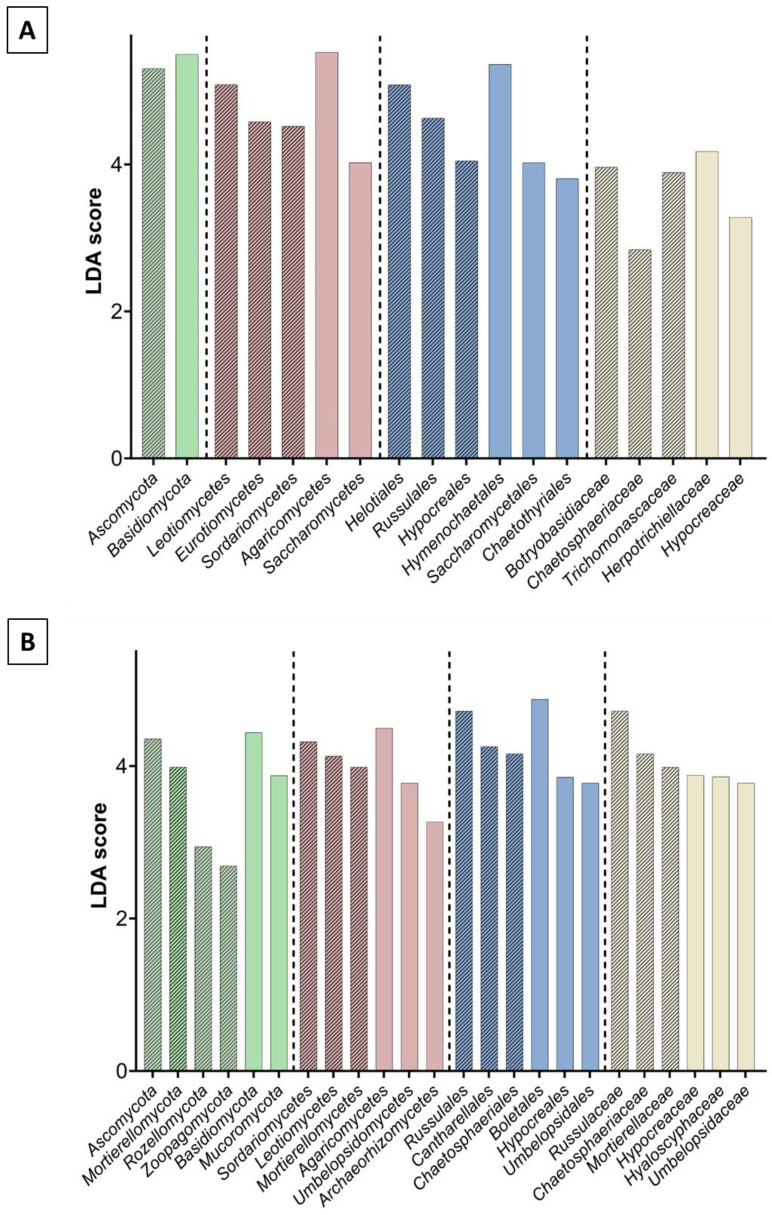
LEfSe analysis showing differences between the metataxonomic results of wood (**A**) and soil (**B**) samples from 1974 (hatched) and 2014 (plain): LDA score representing differences in the abundance of fungal taxa at the phylum (green), class (red), order (blue), and family (yellow) levels.

**Figure 8 biomolecules-13-01466-f008:**
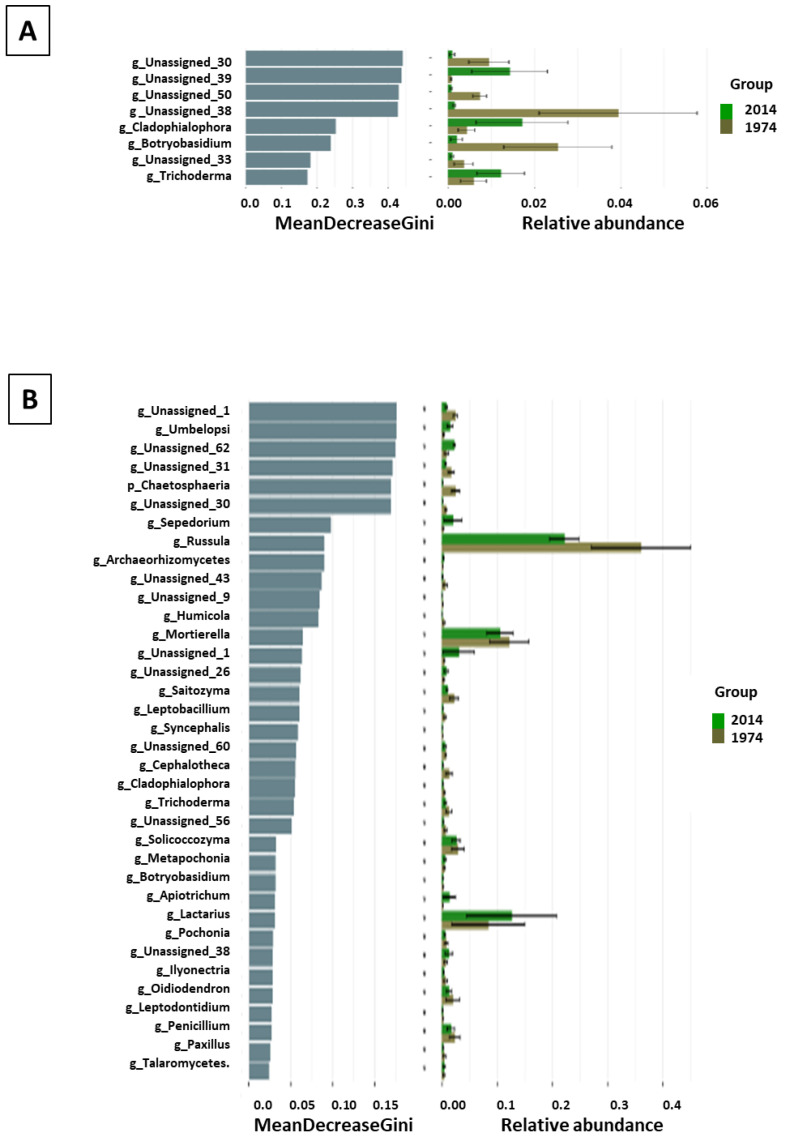
LEfSe analysis showing differences between the metataxonomic results of wood (**A**) and soil (**B**) samples from 1974 and 2014: rf analysis (random forest) showing differences in the abundance of fungal taxa at the genus level.

**Figure 9 biomolecules-13-01466-f009:**
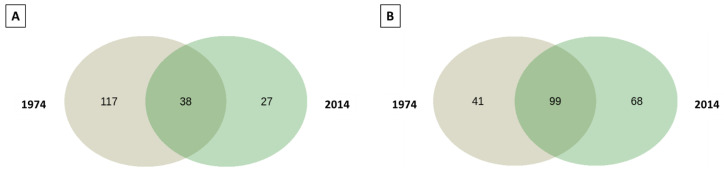
Venn diagram prepared on the basis of ITS1 sequencing of pooled samples of wood (**A**) and soil (**B**) from different years (1974/2014). Data presented in the graph refer to the site-specific unique ASVs and a core microbiome.

**Figure 10 biomolecules-13-01466-f010:**
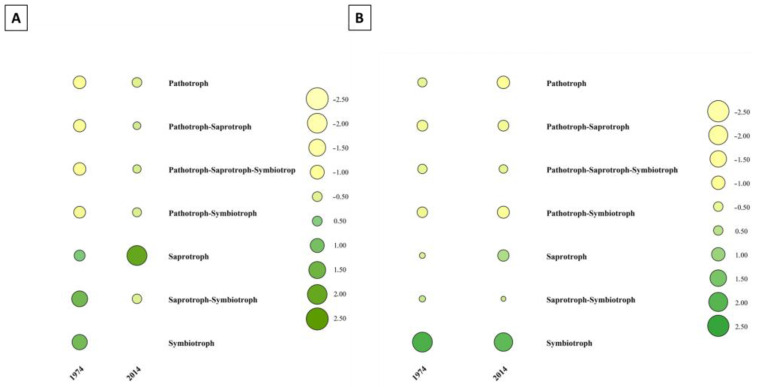
Functional analysis of fungal microbiomes associated with wood (**A**) and soil (**B**) samples performed with FunGuild. To better illustrate the differences, all values were normalized to the scale from +2.5 to −2.5. The size and color of the circle represent different values.

**Figure 11 biomolecules-13-01466-f011:**
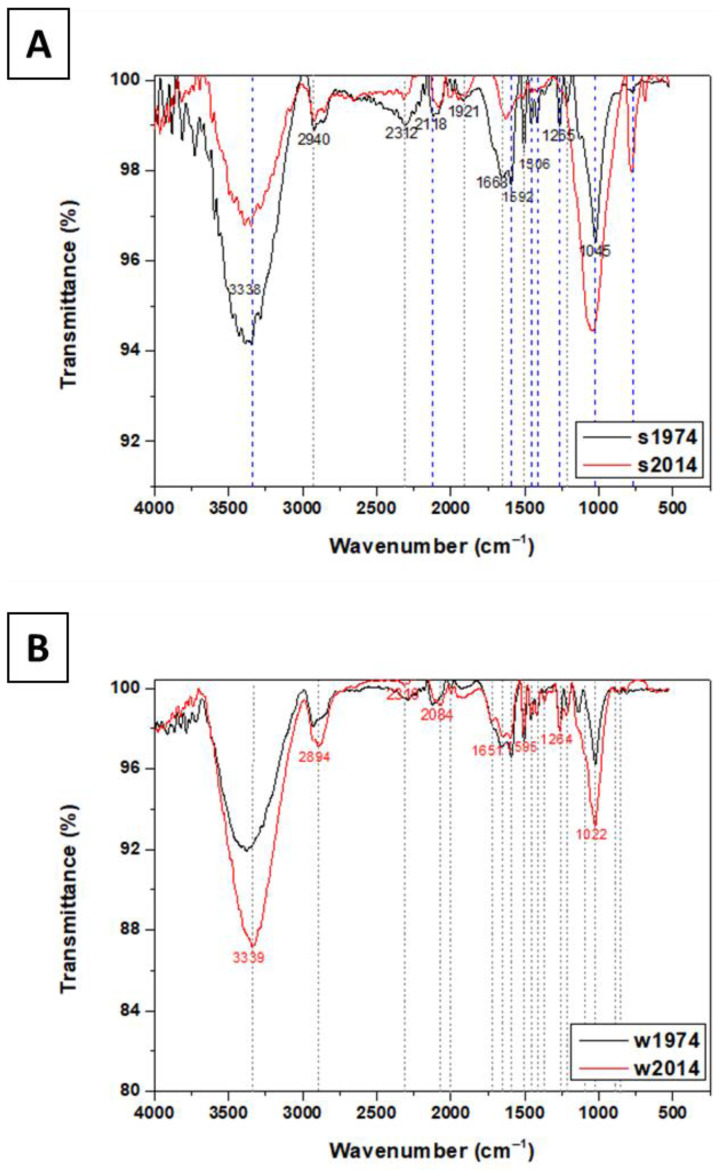
ATR-FTIR analysis of soil (**A**) and wood (**B**) samples taken from logs and nearby.

**Table 1 biomolecules-13-01466-t001:** Percentage content and standard deviation of C, H, N, and O elements in samples of wood (dead since 1974 and 2014) and soil collected in the vicinity of sampled logs.

	1974		2014	
	Wood	Soil	Wood	Soil
N [%]	0.83 ± 0.24	0.94 ± 0.11	0.23 ± 0.10	0.75 ± 0.11
C [%]	51.77 ± 2.79	37.95 ± 14.75	49.60 ± 1.77	13.13 ± 2.76
H [%]	6.29 ± 0.16	4.77 ± 1.77	6.61 ± 0.16	1.90 ± 0.34
O [%]	41.12 ± 2.75	56.33 ± 16.60	43.55 ± 1.60	84.22 ± 3.20

## Data Availability

This Targeted Locus Study project has been deposited at DDBJ/EMBL/GenBank under the accession KHVR00000000. The version described in this paper is the first version, KHVR01000000.

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
