# Peer review of "Metagenomic Analysis of the Composition of Microbial Consortia Involved in Spruce Degradation over Time in Białowieża Natural Forest"

_biomolecules, 2023, doi:10.3390/biom13101466_

Round 1

Reviewer 1 Report

see attached file

Author Response

Dear Professor Uversky,

Thank you for valuable and very helpful comments. We appreciate your help in every step of publishing our paper. I hope that the reviewers will find our explanations and changes satisfactory and the revised manuscript will be acceptable for publication in Biomolecules. For your convenience, we highlighted the new additions and changes in the revised manuscript using “Track-Changes” option in Microsoft Word.

Sincerely,

dr. hab. Grzegorz Janusz

Answers to Reviewers comments

Reviewer 1

Q1. The pretreatment process of the sample is missing. How did wood crush or dissolve before DNA

extraction?

The following information was added into the respective section of the manuscript (Line 128 of the new manuscript).

“Prior to nucleic acid extraction sawdust samples were homogenized to fine powder using a mortar and pestle under liquid nitrogen”

Q2. Add pictures of trees that died in 1974/ 2014 and surrounding soil condition as FIGURE 1. The author discussed that the natural succession of microbial communities occurs according to the intermediate products generated as trees decompose. Visually confirmable difference can make the author's estimation more firm and give readers to substantial application.

As encouraged by the reviewer the pictures were included in the line 105 as Figure 1 and therefore, the numeration of all further figures was adjusted.

Q3. The format of the reference does not conform to the journal regulation.

Thank you for the valuable comment. The references were now formatted with by Endnote according to MDPI style.

Q4. Line 25-29: Shorten the length of the sentence through English correction. Reduce sentence length for background of experiment or explanation of experiment. Fill in the numerical changes that you think are important. Instead of just describing the 2014 and 1974 samples as ‘abstractly different’, describe what has changed through numerical values.

In the lines 25-29 (Abstract) there were two sentences:

“Significantly higher numbers of bacterial taxa were observed in the wood and soil samples from 2014, while the number of fungal taxa was significantly higher in the wood and soil samples from 1974. Most of the bacterial and fungal amplicon sequence variants (ASVs) unique for wood were found in the samples from 1974, while those unique for soil were detected in the samples from 2014.”

However, encouraged by the reviewer we included precise data. New sentence sounds”

“Based on the Lefse analysis more bacterial taxa with were in significantly higher amount were observed in the wood samples from 1974 and soil samples from 2014 (soil: 1974 - 33 taxa, 2014 - 59 taxa; wood: 1974 - 44 taxa, 2014 - 39 taxa), while for fungal taxa it was 1974 (soil: 1974 - 66 taxa, 2014 - 31 taxa; wood: 1974 - 22 taxa, 2014 - 10 taxa). Most of the bacterial and fungal amplicon sequence variants (ASVs) unique for wood were found in the samples from 1974 (117), while those unique for soil were detected in the samples from 2014 (68).”

Q5. Add the GPS location or map showing the sampling location in the preserved area. As far as I know, Google Earth is also acceptable under proper citation in the academic publication.

The GPS location was included in the line 93.

Reviewer 2 Report

The paper by Janusz et al is dedicated to comparison of microbial and fungal diversity in deadwood and respective soil samples in two deadwoods - “born” in 1974 and 2014, using metagenomics approach. It is interesting that higher numbers of bacterial taxa were observed in the wood and soil samples from 2014, while the number of fungal taxa was significantly higher in the wood and soil samples from 1974.  The authors also analysed their chemical composition and revealed the signs of lignocellulose degradation.

This is a clear study that could be of interest to both ecologists and microbiologists. From this point of view, I am slightly surprised that this kind of work  was submitted to “Biomolecules”, it would be rather suitable for Microorganisms or other more profile journals. 

The manuscript is well written and is easy to follow.

Several  minor issues:

  1. Please, provide the primer sequences for ITS1 because there is a confusion - in different papers, the same primers can be called ITS1-2 and ITS3-4. 

  2. Please, briefly discuss the sequence depth, Q30, and what could be the reason for less amount of combined reads for 2014 samples (provided in Table S1). 

  3. To me, Figures 1A, 1C; 5A, 5C would be more illustrative if barplots were used (like in figures 1B, 1D, 5B, 5D) instead of a kind of heatmaps. 

  4. Figures 2 and 6 are very difficult for comprehension. Maybe to try presenting these results somehow else?

Author Response

Dear Professor Uversky,

Thank you for valuable and very helpful comments. We appreciate your help in every step of publishing our paper. I hope that the reviewers will find our explanations and changes satisfactory and the revised manuscript will be acceptable for publication in Biomolecules. For your convenience, we highlighted the new additions and changes in the revised manuscript using “Track-Changes” option in Microsoft Word.

Sincerely,

dr. hab. Grzegorz Janusz

Answers to Reviewers comments

Reviewer 2

Q1. Please provide the primer sequences for ITS1, because there is a confusion – in different papers the same primers can be called ITS1-2 and ITS3-4

The primers sequences for both analyzed amplicons, bacterial V3-V4 region and fungal ITS1, were specified in respective section of manuscript (line 134).

Q2. Please, briefly discuss the sequence depth, %Q30, and what could be the reason for less amount of combined reads in samples from 2014 (as provided in Table S1).

The sequencing depth should be carefully considered during the stage of experimental design. In case of experiment aiming to assess microbial population important factor that should be taken into account is e.g. expected microbial diversity in specific environment. In our analysis we aimed to achieve at least 60K of read pairs for each sample and as specified in table S1 we got even more. In our opinion this sequencing depth was sufficient for the microbial inview experiment and similar with results published elsewhere. With this sequencing depth we obtained numbers of ASV comparable with data presented in literature.

Q30 is a quality parameter score reflecting probability of incorrect base call, which in case of Q30 is 1/1000 (so inferred base call accuracy is 99,9%). It is really hard to speculate about the reasons of slightly decreased Q30 value and less amount of merged reads for a set of soil and wood samples from 2014 subjected for V3-V4 region analysis. It may be simply batch effect of sequencing run, especially that for the same samples analyzed with ITS1 region the respective parameters did not differ substantially from the other. The lower number of merged reads did not significantly affected the number of effective tags used for ASV assignment (other sequencing parameter were also correct) and consequently the downstream analysis of microbial diversity.

As this is only technical aspect of analysis we did not discuss it further in the manuscript.

Q3. To me, Figures 1A, 1C; 5A, 5C would be more illustrative if barplots were used (like in figures 1B, 1D, 5B, 5D) instead of a kind of heatmaps. 

The idea of using heatmaps was suggested by papers on metagenomics published in highly ranked journals. Please see below a few examples:

https://www.mdpi.com/1422-0067/24/12/10042

https://www.mdpi.com/1422-0067/23/16/8978

https://www.nature.com/articles/s41598-022-05858-9

https://microbiomejournal.biomedcentral.com/articles/10.1186/s40168-019-0752-0

Therefore, we prefer to leave these figures treating them as certain standard of presenting this type of results.

Q4. Figures 2 and 6 are very difficult for comprehension. Maybe to try presenting these results somehow else?

Please, concern that in one figure we included taxa abundance influenced by two different factors – time (1974 and 2014) and sample type (wood and soil) in order to allow readers to comprehend complex changes in microbiome composition. Elucidation of this way of presenting results took us long time and we have no idea how to show them in better way, if you don’t mind, we prefer not to change them.

Below there are links to the papers presenting similar results:

https://www.mdpi.com/2311-5637/9/9/827

https://www.mdpi.com/2076-2607/9/2/351